# A biomimetic approach to shielding from ionizing radiation: The case of melanized fungi

Thomas Vasileiou[ORCID]*, Leopold Summerer

ESA - Advanced Concepts Team, European Space Research and Technology Centre (ESTEC), NL-2200AG Noordwijk, Netherlands

* Thomas.Vasileiou@esa.int

## Abstract

Melanized fungi have been shown to thrive in environments with high radionuclide concentrations, which led to the association of the pigment melanin with the protection against ionizing radiation. Several hypotheses regarding the function of melanin have been proposed. Yet, the exact mechanism behind the protective property of melanin is unclear and poorly explored. A better understanding of the mechanisms that are involved in increasing the tolerance of the organisms to ionizing radiation could lead to technology transfer to human-related applications. Effective protection from radiation is essential for human space flight in general and human missions beyond Low Earth Orbit specifically. In this paper, we follow a biomimetic approach: we test two of current hypotheses and discuss how they could be applied to radiation shield designs. First we focus on the interaction of melanin with high energy electrons, which has been suspected to reduce the kinetic energy of the electrons through a cascade of collisions, thus providing physical shielding. Second, we investigate if the spatial arrangement of melanin, organized as a thin film or a collection of hollow microspheres, affects its shielding properties. To this end, we measured experimentally and by numerical simulations the attenuation of $\beta$-radiation as pass through solutions and suspensions of melanin and contrasted the values to the ones of cellulose, a substance with similar elemental composition. Further, we investigate the spatial arrangement hypothesis using Monte Carlo simulations. In agreement with the simulations, our experiments indicated that melanin does not provide improved shielding in comparison to cellulose from $\beta$-radiation. However, our simulations suggest a substantial effect of the spatial arrangement on the shielding performance of melanin, a pathway that could be transferred to the design of composite radiation shields.

**Data Availability Statement:** The data underlying the study is available on the public Zenodo repository (DOI: 10.5281/zenodo.3667494).

**Funding:** The authors received no specific funding for this work.

## Introduction

Protection from ionizing radiation is one of the main challenges for human space flight, considering the biological stress radiation exerts on living organisms [1–4]. Risks associated with radiation exposure are present during the current manned low Earth orbit missions [5] and expected to be a major concern for future exploratory missions on the Moon and Mars [6].

**Competing interests:** The authors have declared that no competing interests exist.

Therefore, new approaches and materials able to address this problem and to provide effective shielding from ionizing radiation are urgently needed.

Environments with high background radiation levels can also be found on Earth. Yet, life is able to persist under these extreme conditions and living organisms have been found to adapt well and in some cases even thrive in such environments (e.g. the Chernobyl disaster site [7, 8]). Studying the mechanisms these organisms cope with ionizing radiation, may give inspiration for novel radiation shielding practises and materials. Interestingly, the percentage of fungi that synthesize the pigment melanin is substantially higher in the high background radiation areas than what is normally observed in places with typical ionizing radiation levels [8, 9]. This observation has lead to the hypothesis that melanin may provide a survival advantage in environments with ionizing radiation and has fueled many studies—from which we summarize the most relevant results in the following—confirming that indeed melanin plays a role in how fungi endure the ionizing radiation exposure.

Specifically, it have been demonstrated that melanized fungi exhibit increased proliferation when exposed to ionizing radiation in comparison to non-melanized strains or to low ionizing radiation environments [8, 10, 11]. Additionally, the protective effect of melanin can be transferred to organisms that do not produce the pigment; non-melanized fungal cells exhibited higher survival rates after irradiation, when melanin extracted from melanized cells was added to the culture medium [12]. The transfer of the protective effects of melanin has been also demonstrated on mice models; intravenous injection [13–15] or ingestion [16] of melanin resulted in higher survival rates after exposure to $\gamma$-radiation in comparison to controls. Recently, the protective role of melanin was also demonstrated in fungal cells for deuteron radiation [17].

Still, how melanin is involved in the protection against the effects of ionizing radiation is poorly understood. The proposed mechanisms range from physical processes, like direct interaction with high-energy photons and electrons, to biochemical effects, like quenching of the cytotoxic free radical produced by radiation. Supporting the physical interaction hypothesis is the fact that fungal cells deposit melanin as a layer on the inner side of the cell membrane [18, 19], indicating that this layer might provide some kind of shielding. The word *shielding* is used here to indicate the physical interaction between radiation and melanin, even if the latter resides inside the cell. Equivalent shielding behaviour from cells has already been observed in the case of ultraviolet (UV) radiation: zebra-fish larvae employ an umbrella of melanocytes to protect the sensitive haematopoietic niche from UV-radiation [20] and human keratinocytes uptake synthetic melanin nano-particles and form a supranuclear cap to lessen UV damage [21]. While the UV absorption spectrum of melanin is well documented, the interaction of melanin with high-energy particles has so far only been hypothesized.

More specifically, it has been proposed that the $\pi$-electron rich oligomer units that compose melanin dissipate the energy of incident electrons in a controlled way [22]. In addition, it has been suggested that the interaction of melanin with photons through the Compton scattering mechanism—the inelastic scattering of a photon by charged particles—attenuates the photon energy and produces secondary electrons which melanin traps [10, 16]. The physical interactions also supported by the speculations that some fungi are able to utilize ionizing radiation as an energy source for metabolic processes, an ability termed *radiotrophism*. Radiotrophism was initially proposed by Zhdanova et al., after reporting hyphal growth of various fungi towards sources of radioactivity [7]. The implication of melanin in radiotrophism was suggested by Dadachova et al. [10], after demonstrating that ionizing radiation changes the electronic properties of melanin and that melanized fungi incorporate acetate faster under radiation exposure. We remark that acetate accumulation is indicative of the heterotrophic capability in photosynthetic bacteria. Two more studies provide support to the radiotrophism hypothesis: exposure

to ionizing radiation reduces the adenosine triphosphate (ATP) levels only in melanized fungal cells, resembling utilization of ATP during the stage of simple sugar composition in photosynthesis [23]. In addition, significant up-regulation of ribosomal biogenesis genes has been reported in melanized yeast in carbon limited media after irradiation in comparison to melanin-deficient mutant [24], indicating that melanin may be able to harness the energy needed by the ribosomal biogenesis machinery.

Another hypothesis relates the spatial arrangement of melanin inside the fungal cells with its protective properties. In the case of the pathogenic fungus *Cryptococcus neoformans*, melanin is arranged in a spherical shape covering the inner surface of the cell membrane [18]. This arrangement has been suggested to increase the scattering of incident photons resulting in superior shielding [12]. Melanin hollow particles of roughly spherical shape, also referred to as "ghosts", can be extracted from *C. neoformans* cells by digestion with acid, a process which leaves the original melanin structure intact [12, 18]. Irradiation of non-melanized fungal cells with a [137]Cs source in the presence of intact *C. neoformans* ghosts resulted in higher survival rates than in the presence of crushed ghosts [12]. Moreover, higher attenuation of X-rays has been recorded for suspensions of *C. neoformans* ghosts in comparison to *Sepia officinalis* melanin, which forms smaller nano-particles [12, 25].

In the present study, we draw inspiration from the biological studies on melanized fungi and examine possible ways to transfer these principles into the design of shielding material for ionizing radiation. First, we test the potential of melanin as a shielding material against $\beta$-radiation and we examine the proposed role of melanin in "trapping" secondary Compton electrons. In contrast to previous studies, we separate the physical from any chemical or biological effects and we compare, experimentally and through numerical simulations, the transmitted electron energy through materials that contain or are free of melanin. In a second step, we consider the spatial arrangement hypothesis; we use numerical simulations to understand how the arrangement of different materials in a composite radiation shield affects its performance.

## Materials and methods

### Chemicals and sample preparation

We obtained the following chemical from Merck: melanin synthetic (M8631), melanin from *S. officinalis* (M2649) and ammonia solution (25 wt. %). We purchased cellulose nano-crystals (CNC, 12 wt. % aqueous gel) from the University of Maine, USA.

We prepared mixtures of synthetic and *S. officinalis* melanin in DI water and 2 mol L$^{-1}$ ammonia solution. We note that the synthetic melanin is soluble to the ammonia solution, whereas all other combinations resulted in suspensions. We started by adding 20 mg of melanin in 1.14 ml of the solvent. In case of the solution, we vortex mixed for 1 min, we allowed for any undissolved particles to sink and we used the total volume of the supernatant for the experiments. For the suspensions, we added the ingredients in standard 15 ml falcon tubes and treated them in an ultrasonic bath (Branson 2510E-DTH, 100 W, 42 kHz) for 30 min.

We prepared CNC solution by diluting 167 mg of 12 wt. % CNC aqueous gel with 0.993 mL DI water, resulting in the same weight fraction as the melanin mixtures. Prior to the shielding experiments, we transferred all of the samples in the standard 12-well microtiter cell culture plates covered by a thin low-density poly-ethylene film to minimize evaporation.

### Shielding experiments

We determined the shielding properties of samples using the following procedure; we placed the sample to be tested between the radioactive source and the detector (semiconductor spectrometer Cube 527, quasi-hemispherical 500 mm$^3$ CdZnTe crystal, GBS Elektronik GmbH,

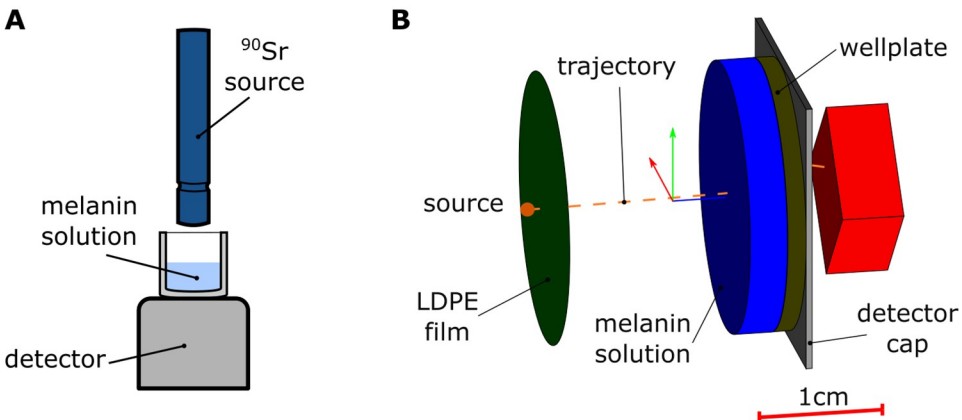

**Fig 1. Setup schematic and simulation geometry.** (A) Illustration of the shielding experiments. (B) Simulated geometry for the shielding experiments in Geant4.

Germany), in a vertical configuration as shown in Fig 1A. We irradiated the sample for 1000 sec with the $^{90}$Sr $\beta$-source. The specific source was selected because its energy spectrum overlaps with the Compton electron spectrum previous studies suggest an increased shielding effect is present (a detailed justification is given in S1 Text and a comparison of the spectra is plotted in S1 Fig). In all cases, the bottom of the well was in contact with the detector and we placed the source at the center of the well just above its edge. The detector was calibrated to the energy range of 10 keV to 2071 keV, which was separated into $K = 1020$ equally spaced bins. We remark that 10 keV is the minimum detectable energy. The bin counts were recorded by the WinSpec-I software (GBS Elektronik GmbH, Germany). Therefore, from each experiment we obtain a histogram of the deposited energy spectrum on the detector, which we describe by the energy mid-value of the bin, $E_{b,i}$, and the number of counts of the bin, $u_i$, with $i = 1, 2, \ldots, K$. We refer to the total number of counts, $\sum_{i=1}^{K} u_i$, as the histogram content. We estimated the absorbed dose at the detector by adding up all the energy contributions:

$$D = \frac{1}{m_d} \sum_{i=1}^{K} u_i E_{b,i} \tag{1}$$

where $m_d = 2.9$ g is the mass of the detector crystal.

To better assess the effect of melanin, we tested all samples against melanin-free controls, specifically wells with the same volume of solvent. In the following, we report the relative absorbed dose,

$$RD = \frac{D}{D_c} \tag{2}$$

where $D_c$ is the absorbed dose in the case of control. For no relative improvement to shielding $RD \to 1$, whereas for perfect shielding $RD \to 0$. The introduction of $RD$ serves an alternative role; when determining $D$ experimentally, there is an additional variability in the measurements of samples of same composition at different experimental campaigns, in between of which the radioactive source has to be unmounted from the setup for safe keeping. The variability can be attributed to minor changes on the geometry of the setup. The use of $RD$ remedies the situation and allows to compare measurements from difference experimental campaigns, given that $D$ and $D_c$ were acquired during the same campaign.

## Spatial arrangement

The original shielding experiments that support the spatial arrangement hypothesis were performed by Dadachova et al. [12] and compared the attenuation of X-rays between suspensions of *C. neoformans* ghosts and *S. officinalis* nanoparticles. We expand on the previous idea and we simulated a wider range of arrangements with the $^{90}$Sr and X-ray sources. Stated differently, we investigate if there is an optimal arrangement for a composite shield, made out of two materials with given atomic numbers, $Z$. To distinguish between the two materials, we loosely refer to them as low- and high-$Z$ material. We characterize the composite shield the material volume ratio, $R_V$, and the areal density, $\rho_A$; we define $R_V$ as the ratio of the volume of high-$Z$ material over the total volume of the shield and $\rho_A$ as the mass of the high-$Z$ material per unit area of the shield.

For the simulations, we fix the values for $\rho_A$ and $R_V$, resulting in composite shields with constant mass and constant height per unit area for all possible arrangements. We simulated composite materials of two configurations: layered and mixture. In the layered configuration, which we referred to as "film", the high-$Z$ material forms a single sheet, which is either sandwiched between the low-$Z$ material or placed at the outer facet of the composite, as shown in Fig 2A. The exact placement is described the relative position parameter $h_r \in [0, 1]$. The mixture configuration is constructed by the repetition of a unit cell, a cube with side $a$ of the low-$Z$ material containing the high-$Z$ material in the three arrangements shown in Fig 2B. In the "sphere" arrangement the high-$Z$ forms a sphere with its center coinciding with the center of the unit cell. In the "packed sphere" and "ghost" configurations, the high-$Z$ material is placed in a body-centered cubic arrangement (borrowing the term from atomic crystal characterization) of solid and hollow spheres, respectively. For the "ghost" configuration, the outer radius is equal to $\sqrt{3}a/4$ and the inner radius is defined by the $R_V$ parameter. To facilitate the comparison, we parameterized the lattice geometries by the equivalent radius, $R_{eq}$, the radius of a sphere with volume equal to the total volume of the high-$Z$ material contained in the unit cell.

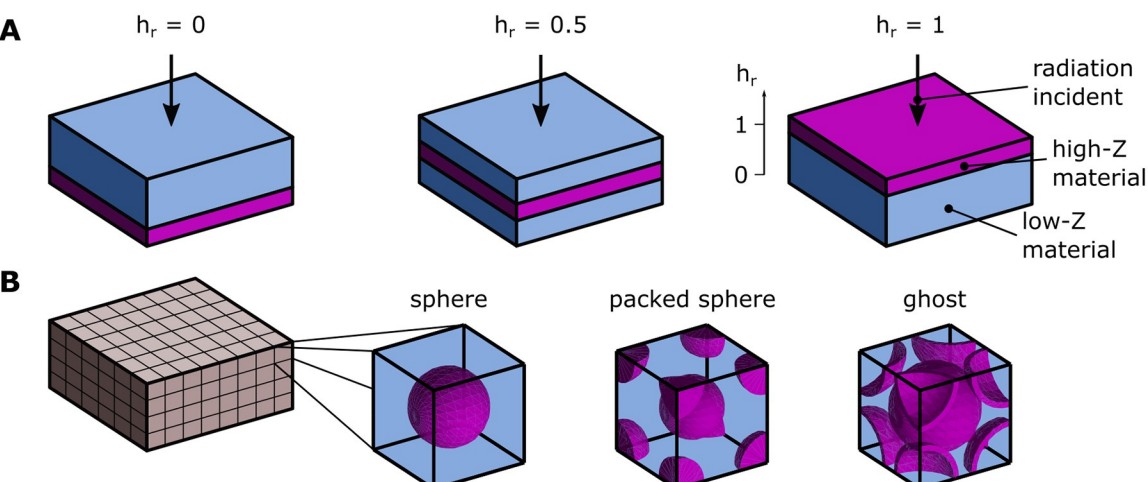

**Fig 2. Schematic of the simulated spatial arrangements.** (A) Illustrations of the film spatial arrangement, at three relative positions ($h_r$ = 0, 0.5 and 1). The direction of the incoming radiation is indicated by the arrow. (B) Illustration of the lattice spatial arrangement for three configurations: sphere, packed sphere and ghost.

We quantify the effectiveness of the difference arrangements by comparing the radiant fluence

$$H = \frac{1}{A_s} \sum_{i=1}^{M} E_{t,i} \qquad (3)$$

after the shield, where $M$ and $E_{t,i}$ is the total number and the energy of the transmitted particles and $A_s$ is the shield area. For consistency, we report the relative radiant fluence, $RH = HH_c^{-1}$, which is normalized by the fluence $H_c$ of the control sample. We use as control a homogeneous shield made of an ideal mixture of the two materials. Specifically, the density of the mixture is $\rho_m = R_V \rho_h + (1 - R_V)\rho_l$, where $\rho_h$ and $\rho_l$ are the densities of the high-$Z$ and low-$Z$ components respectively. The weight fraction is equal to $R_V \rho_h \rho_m^{-1}$ for the high-$Z$ and $(1 - R_V)\rho_l \rho_m^{-1}$ for the low-$Z$ material.

## Numerical simulations

We performed Monte Carlo simulations using the software Geant4 [26, 27], to gain insight on the theoretical values for $H$ and $D$. All simulations were performed using the low-energy electromagnetic model (Livermore library) of Geant4 [28], with validity down to 250 eV.

For comparing with the shielding experiments, we simulated the exact same conditions and geometry, as shown Fig 1B. To avoid any systematic error, we model the $^{90}$Sr $\beta$-source using the energy spectrum taken from [29], which is plotted in panel (A) of S2 Fig. The source emits electrons at a right cone towards the sample, with apex semi-angle of 15.2˚. For the simulations, we neglected the contribution from the infrequent $\gamma$-decay of $^{90}$Y. We modelled the solution as simple mixtures of elements, with the following simplifying assumption; the addition of melanin does not change the volume of the mixture (same volume as the solvent), but we added the contribution of the mass of the melanin to the density of the solution. For the suspensions we use the same modeling, since the melanin particles did not had the time to precipitate between the sonication and irradiation. In terms of the geometry, we model the sample and the detector (only the semiconductor crystal and the aluminum cap) and we determine the deposited energy, $E_d$, on the semiconductor crystal. We simulated $5 \times 10^6$ primary events and we summed up the contributions from the primary and all the secondaries particles in $E_d$. We discard events with $E_d < 10$ keV, which resides outside the range of our detector. The simulated dose was computed as

$$D = \frac{1}{m_d} \sum_{i=1}^{P} E_{d,i} \qquad (4)$$

where $P$ is the total number of events recorded by the detector. More information about the settings, geometries and the composition of materials used in the simulations are given in S1 Text.

The simulations to assess how the arrangement of different material affects the shielding properties of the composite were performed in a similar manner; in this case though we recorded the transmitted energy through the composite. We used melanin (synthetic, see S1 Table for elemental composition) and water as the high-$Z$ and low-$Z$ materials respectively. We fixed the geometric parameters of the composite shield at $\rho_A = 3.37$ mg cm$^{-2}$ and $R_V = 0.234$. We simulated various configuration for $h_r$ (ranging from 0 to 1) and $R_{eq}$ (ranging from 16 nm to 8192 nm) with the $^{90}$Sr and the 40 kVp X-ray source. The energy spectrum of the X-ray photons was calculated using the method of interpolating polynomials for a tungsten anode source without filter, as described in [30]. The resulting spectrum is shown in panel (B) of S2 Fig.

## Statistical analysis

The dose accumulation and the radiant fluence can be modelled as a compound Poisson process: $D$ and $H$ are the sum of independent and identically distributed (i.i.d.) random variables, where the numbers of terms to be added follows the Poisson distribution. In general, the energy per particle, $E_{p,i}$, which is either deposited on the detector or exiting the shield, is the i.i.d. random variable and the number of incident particles in a given time interval follows Pois ($\lambda$), with $\lambda$ the event rate. The probability distribution of $E_{p,i}$ derives from the distorted spectrum of the radiation source, as it passes through the shielding material. The parameter $\lambda$ is related to the decay rate of the radioactive source, but it is also affected by the shielding material through the absorption and generation of particles.

In the case of the shielding experiments, the theoretical mean and the variance of $D$, denoted by $\mu_D$ and $\sigma_D^2$ respectively, are given by $\mu_D = m_d^{-1}\lambda\mathbf{E}[E_{p,i}]$ and $\sigma_D^2 = m_d^{-2}\lambda\mathbf{E}[E_{p,i}^2]$, where $\mathbf{E}$ denotes the expected value. Therefore, Eqs (1) and (4) constitute appropriate estimators of $\mu_D$, the first one in case the histogram of the energy spectrum is available, whereas the second if each of the individual realizations of $E_{p,i}$ are registered. The respectful estimators for $\sigma_D^2$ in case of histograms and individual recording are

$$s_D^2 = \frac{1}{m_d^2}\sum_{i=1}^{K} u_i E_{b,i}^2 \tag{5}$$

$$s_D^2 = \frac{1}{m_d^2}\sum_{i=1}^{P} E_{d,i}^2. \tag{6}$$

These estimators also indicate that doses obtained from two separate experiments can be combined by averaging the means and the variances of the two. As $P \to \infty$, $D \sim \mathcal{N}(\mu_D, \sigma_D^2)$ as a consequence to the Berry–Esseen theorem [31]. For our calculation of $D$, the individual terms in the summations are in the order of $10^6$, thus we treat $D$ as normally distributed. We apply similar reasoning and treatment for the calculation of $H$.

The shielding material can change both the energy spectrum and the rate of the incident particles on the detector. Comparison between different shielding approaches is performed in three steps: first we employ the Anderson–Darling test [32, 33] to detect differences in the energy spectra, then we apply the binomial test to compare the rate of the two event or the content of the two histograms [33]. Finally we apply the Z-test to compare the dose. In all of the previous tests, the null hypothesis states that the random variables were drawn from the same distribution. We accompany the $p$-values from the hypothesis tests with estimated effect sizes. To quantify differences between two energy spectra we calculate the Kolmogorov–Smirnov score, $D_{KS}$, namely the maximum absolute difference between the empirical cumulative distribution functions computed from the histogram data [33]. For the rate comparisons, we estimate Cohen's $w$ [34] and we report the Z-test score

$$\theta = \frac{(D_1 - D_2)}{\sqrt{s_{D_1}^2 + s_{D_2}^2}} \tag{7}$$

for the dose comparisons, where the subscripts 1 and 2 indicate the samples to be compared.

In the case of $RD$ and $RH$, we follow a similar procedure. As random variables, $RD$ and $RH$ have ratio distribution as the quotient of two normally distributed random variables. A normal approximation of the ratio distribution is valid under some assumptions [35, theorem1]. Specifically, the normal approximation can be justified if both variables in the ratio are strictly

positive and the root square sum of their inverse coefficient of variation is smaller than 0.1, which translates to $\sqrt{\mu_D^2\sigma_D^{-2} + \mu_{D_c}^2\sigma_{D_c}^{-2}} < 0.1$ in the case of $D$. The variables $\mu_{D_c}$ and $\sigma_{D_c}$ denote the mean and standard deviation of $D_c$. In the case of $RD$, the mean of the approximation is given by Eq (2) and the standard deviation by $RD\sqrt{\sigma_D^2\mu_D^{-2} + \sigma_{D_c}^2\mu_{D_c}^{-2}}$ [35]. These criteria were always valid in our case, therefore we use the Z-test to detect differences in the values of $RD$ and $RH$. Moreover, we assume that histograms of the same sample that recorded during different experimental campaigns differ mainly on the content and not on the shape. This implies that the shape of histograms between different samples can be compared by the Anderson–Darling test. For the content though, we apply Fisher's exact test under the null hypothesis that the ratio of the histogram content between sample and control remains constant. For Fisher's exact test, the associated Cohen's $w$ effect size is reported.

For the comparison of multiple experiments, first we assess the hypothesis that all incident particles have the same energy spectrum using the k-sample Anderson–Darling test. To compare the rates, in the case of $D$ and $H$ we use Pearson's $\chi^2$-test for goodness of fit under the null hypothesis that the categorical data has the same frequency; each separate experiment constitutes a category and the histogram content the corresponding frequency. In the case of $RD$ and $RH$, we use $\chi^2$-test for independence, where the content of the control histogram is also used. In both cases, we report Cohen's $w$. If any of these two tests reject the null hypothesis, we perform pairwise comparisons, similar to the case of two samples. The hypothesis rejection rates for the pairwise comparisons are adjusted using the Bonferroni correction.

In the following, all quantities are reported as the expected value with the 99% confidence interval (CI). The $p$-values correspond to two-sided tests and are reported as continuous variables. We indicate $p$-values smaller than 0.001 as $<$0.001. We signify rejection of the null hypothesis with one, two or three asterisks at confidence level of $\alpha = 0.05$, 0.01 and 0.001 respectively. We note that for the k-sample Anderson–Darling, the provided interpolating formula for the $p$-values range from 0.001 to 0.25 and as a result the reported values are restricted to this range.

## Results

### Melanin shielding from $\beta$-radiation

The experimentally measured values for $RD$ for the different mixtures are shown in Fig 3A, along with the numerically simulated values. All samples exhibit measurable $RD$, namely the deposited energy on the detector was substantially lower in comparison to control. An example of a spectrum recorded during the experiments is shown in Fig 3B for a *S. officinalis* melanin and water suspension, where it is contrasted to the simulated one. A comparison of the previous spectrum to its control is shown in S3 Fig. All spectra are normalized to represent the estimated probability density function (PDF) for better inspection (area under the curve is equal to 1). In comparison to the simulated one, the recorded spectrum from the experiments has more content on the low energy region; nonetheless, the simulations captured the overall trend and the discrepancy is attributed to neglected detector dynamics, as discussed in detail in S1 Text and S4 Fig.

Statistical comparison among the experiment recordings revealed difference in both the energy spectra and the rate of incident particles (k-sample Anderson–Darling $p < 0.001$, $\chi^2$-test $p < 0.001$, Cohen's $w = 0.0094$). Similar results were obtained from comparing the simulated dose (k-sample Anderson–Darling $p < 0.001$, $\chi^2$-test $p < 0.001$, Cohen's $w = 0.0193$). Differences were also detected among the experiment controls (k-sample Anderson–Darling $p < 0.001$, $\chi^2$-test $p < 0.001$, Cohen's $w = 0.1073$). The pairwise comparisons for the

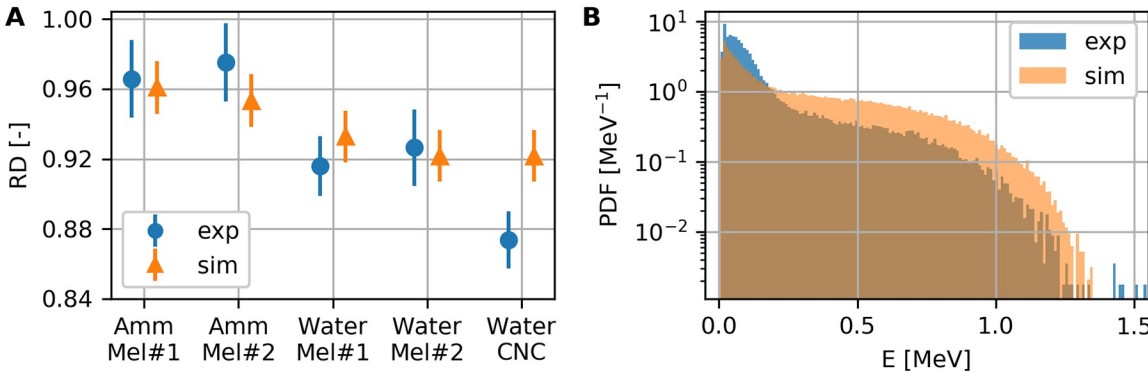

**Fig 3. Shielding effectiveness of melanin mixtures.** (A) Shielding effectiveness of melanin mixtures in ammonia (Amm) and water, for synthetic melanin (Mel#1) and melanin of *S. officinalis* (Mel#2). For comparison, slurry of cellulose nano-crystals (CNC) was also tested. Experimental measurements are shown in circles and numerical simulation results in squares. (B) Comparison between experiment and simulation of the estimated probability density function of the deposited energy spectrum. Results for the *S. officinalis* and water suspension.

experimental and simulation data are presented in Table 1. The correspondence between samples and controls, along with the experimental campaign they were recorded in, is summarized in S2 Table. We point out that the reported comparisons are with respect to *RD* in the case of the experiments and to *D* in the case of simulations and experiment controls.

For the melanin mixtures, we detected no noticeable influence of the melanin type (synthetic or *S. officinalis*) on *RD*, but the observed differences seem to be connected to the solvent. The measured *RD* for CNC solution is lower than the ones for the water-melanin suspensions, although the simulated value show no detectable discrepancy. For the experimental data, the variations in *RD* seem to be associated to the deposited energy spectrum, whereas for the simulations the variations seem to stem from the histogram content. In particular, the experimental values of $D_{KS}$ are higher than the simulated ones for comparisons between the same samples, whereas Cohen's *w* follows the opposite trend. The highest values for $\theta$ were recorded among the experiment controls, confirming that direct comparison of *D* between different experimental campaigns is not straightforward. In accordance to our assumption, histograms recorded during different experimental campaigns (Ammonia#1 vs Water#1 and Water#1 vs Water#2) have an order of magnitude higher values for Cohen's *w* than histograms recorded during the same campaigns (Ammonia#1 vs Water#2), while the values of $D_{KS}$ are comparable for all sample comparisons.

## Spatial arrangement of melanin affects shielding effectiveness

Fig 4A presents the simulated *RH* for the [90]Sr source. The results for the X-ray source are shown in S5 Fig, because *RH* was identical for all arrangements within the estimated uncertainties. To amplify the effect of the spatial arrangement, we repeated the simulations with tungsten (W, high-*Z*) and poly-ethylene (PE, low-*Z*) instead, two materials that have been proposed as components for radiation shields [36]. For the W-PE composite, $R_V$ was kept as previously and $\rho_A = 38.71$ mg cm$^{-2}$, resulting in a shield with same height per unit area as in the melanin-water case. The results of the simulations are presented in Fig 4B and 4C.

For the melanin-water composite and the [90]Sr source, the film configuration showed noticeable trend between *RH* and $h_r$, whereas among the lattice configurations the effect was negligible. The most favorable configuration with respect to shielding is the film with $h_r = 0$. The results for the W-PE composite with the [90]Sr source reveal the tendency of *RH* to decrease

**Table 1. Pairwise comparisons for shielding experiments.**

| | $D_{KS}$ | $p$(AD) | Cohen's $w$ | $p$(HC) | $\theta$ | $p$(Z) |
|---|---|---|---|---|---|---|
| | | | **Experiments** | | | |
| Amm Mel#1 vs Amm Mel#2 | 0.0048 | 0.250 | 0.0036 | 0.075 | −0.78 | 0.437 |
| Amm Mel#1 vs Water Mel#1 | 0.0165 | 0.001** | 0.0048 | 0.008 | 4.59 | < 0.001*** |
| Amm Mel#1 vs Water Mel#2 | 0.0214 | 0.001** | 0.0034 | 0.102 | 3.26 | 0.001* |
| Amm Mel#1 vs Water CNC | 0.0235 | 0.001** | 0.0099 | < 0.001*** | 8.61 | < 0.001*** |
| Amm Mel#2 vs Water Mel#1 | 0.0170 | 0.001** | 0.0083 | < 0.001*** | 5.44 | < 0.001*** |
| Amm Mel#2 vs Water Mel#2 | 0.0216 | 0.001** | 0.0070 | 0.001** | 4.03 | < 0.001*** |
| Amm Mel#2 vs Water CNC | 0.0239 | 0.001** | 0.0135 | < 0.001*** | 9.46 | < 0.001*** |
| Water Mel#1 vs Water Mel#2 | 0.0058 | 0.103 | 0.0015 | 0.422 | −0.98 | 0.329 |
| Water Mel#1 vs Water CNC | 0.0092 | 0.001** | 0.0053 | 0.001* | 4.61 | < 0.001*** |
| Water Mel#2 vs Water CNC | 0.0058 | 0.074 | 0.0066 | 0.001** | 4.98 | < 0.001*** |
| | | | **Simulations** | | | |
| Amm Mel#1 vs Amm Mel#2 | 0.0044 | 0.250 | 0.0014 | 0.552 | 1.30 | 0.194 |
| Amm Mel#1 vs Water Mel#1 | 0.0069 | 0.006 | 0.0177 | < 0.001*** | 7.72 | < 0.001*** |
| Amm Mel#1 vs Water Mel#2 | 0.0112 | 0.001** | 0.0190 | < 0.001*** | 9.63 | < 0.001*** |
| Amm Mel#1 vs Water CNC | 0.0093 | 0.001** | 0.0231 | < 0.001*** | 9.59 | < 0.001*** |
| Amm Mel#2 vs Water Mel#1 | 0.0041 | 0.163 | 0.0164 | < 0.001*** | 6.42 | < 0.001*** |
| Amm Mel#2 vs Water Mel#2 | 0.0078 | 0.001** | 0.0176 | < 0.001*** | 8.33 | < 0.001*** |
| Amm Mel#2 vs Water CNC | 0.0082 | 0.008 | 0.0217 | < 0.001*** | 8.30 | < 0.001*** |
| Water Mel#1 vs Water Mel#2 | 0.0058 | 0.032 | 0.0012 | 0.597 | 1.91 | 0.057 |
| Water Mel#1 vs Water CNC | 0.0052 | 0.197 | 0.0054 | 0.022 | 1.89 | 0.059 |
| Water Mel#2 vs Water CNC | 0.0050 | 0.142 | 0.0041 | 0.079 | −0.02 | 0.987 |
| | | | **Experiment controls** | | | |
| Ammonia#1 vs Water#1 | 0.0043 | 0.250 | 0.1257 | < 0.001*** | 31.31 | < 0.001*** |
| Ammonia#1 vs Water#2 | 0.0098 | 0.001** | 0.0177 | < 0.001*** | 7.31 | < 0.001*** |
| Water#1 vs Water#2 | 0.0090 | 0.001** | 0.1082 | < 0.001*** | −23.47 | < 0.001*** |

$D_{KS}$: Kolmogorov–Smirnov effect size, $p$(AD): Anderson–Darling test $p$-value, $p$(HC): histogram content test $p$-value (binomial in case of $D$, Fisher's exact test in case of $RD$), $\theta$: Z-test score, $p$(Z): Z-test $p$-value.

Asterisks signify rejection of null hypothesis at $p$-value:

* < 0.05%,

** < 0.01% and

*** < 0.001%.

with decreasing $h_r$ and $R_{eq}$. Surprisingly, for the X-ray source the different configurations resulted in well separated regions. In this case, the sphere configuration demonstrated $RH > 1$, practically performing worst that the ideal mixture of the two material. The diminished performance of the sphere configuration can be attributed to the fact that for perpendicular incidence of the incoming photons, there are direct trajectories through the material that do not intersect with the high-$Z$ material. Overall, the film configuration with $h_r = 0$ followed by the ghost with $R_{eq} \approx 1$ µm appear to provide the highest shielding in all cases.

The absence of detectable differences in $RH$ for melanin-water composite with the X-ray source is not in agreement with the shielding results from the Dadachova et al. experiments. Consequently, we examined if the discrepancy can be attributed to differences in $\rho_A$. In the original experiments, three different amounts of melanin were added to 96-well plates, in particular 30 mg, 50 mg and 100 mg. For a typical well diameter of 6.94 mm for the 96-well plate, $\rho_A$ is roughly equal to 82 g cm$^{-2}$, 137 g cm$^{-2}$ and 274 g cm$^{-2}$. We repeated the simulations for

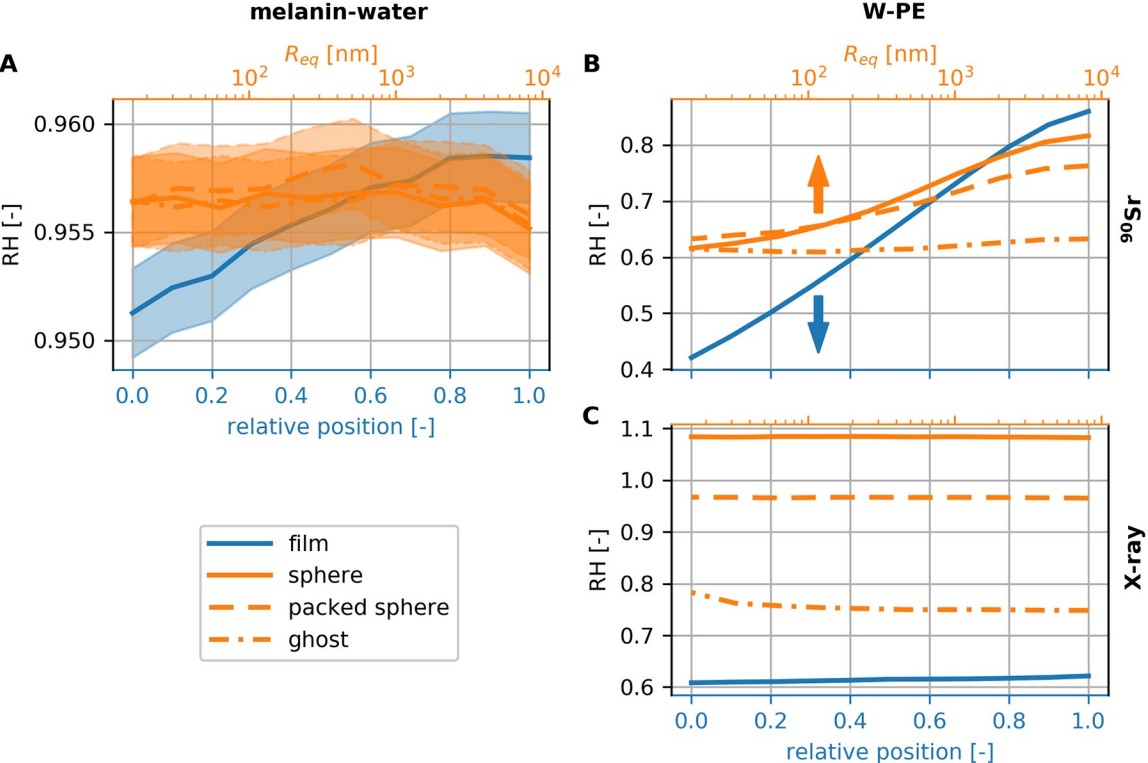

**Fig 4. Comparison between different arrangements.** (A) Relative radiant fluence for different spatial arrangements for the melanin-water composite and the [90]Sr source. The relative film position is marked on the bottom *x*-axis and the equivalent radius for the lattice configurations at the top logarithmic *x*-axis. (B) Relative radiant fluence for the W-PE composite and the [90]Sr source. The CI intervals are smaller than the line width. (C) Relative radiant fluence for the W-PE composite and the 40 kVp X-ray source. The CI intervals are smaller than the line width.

the aforementioned values of $\rho_A$ for two configurations: melanin ghosts and film with $h_r = 0$. We model ghosts as perfect hollow spheres with an outer diameter of 3 mm and wall thickness of 200 nm, values that closely resemble *C. neoformans* ghosts [18]. We assume that the ghosts are closely packed with the in-between space filled with water, as in Fig 2B, resulting in $R_V = 0.234$. Table 2 holds the simulated values for *RH* for the two arrangements, marked with the subscripts *f* for the film and *g* for the ghosts, along with *p*-values from the statistical testing and the estimated effect sizes. In accordance to the previous results, we did not observe any detectable differences in the *RH* values, the histogram shapes or the histogram content for the X-ray source. In contrary, the ghost arrangement has a substantially lower *RH* for the [90]Sr source up to $\rho_A$ of 137 mg cm$^{-2}$, but not for $\rho_A = 274$ mg cm$^{-2}$. The difference in *RH* values seems to arise from the number of incoming particles on the detector and not from the shape of the energy distribution; $\theta$ shows stronger correlation to Cohen's *w* than to $D_{KS}$.

## Discussion

Our study investigated possible bioinspired mechanisms to improve current radiation shielding techniques. In addition, through this study we test some of the main speculation regarding the high radiation resistance of the melanized fungus *C. neoformans*. With respect to interaction of melanin with *β*-radiation, our experiments indicate that melanin does not exhibit improved shielding; the recorded *RD* for melanin was comparable to cellulose, a substance

**Table 2. Simulation results for spatial arrangement.**

| $\rho_A$ [mg cm$^{-2}$] | $RH_g$ | $RH_f$ | $D_{KS}$ | $p$(AD) | Cohen's $w$ | $p$(HC) | $\theta$ | $p$(Z) |
|---|---|---|---|---|---|---|---|---|
| | | | | X-ray source | | | | |
| 82 | 0.946±0.002 | 0.946±0.002 | 0.0004 | 0.250 | 0.0004 | 0.289 | −1.04 | 0.299 |
| 137 | 0.912±0.002 | 0.913±0.002 | 0.0011 | 0.250 | 0.0002 | 0.577 | −1.00 | 0.318 |
| 274 | 0.837±0.002 | 0.838±0.002 | 0.0017 | 0.250 | 0.0003 | 0.465 | −1.52 | 0.129 |
| | | | | $^{90}$Sr source | | | | |
| 82 | 0.475±0.002 | 0.496±0.002 | 0.0065 | 0.017* | 0.0251 | < 0.001*** | −29.09 | < 0.001*** |
| 137 | 0.256 0.001 | 0.268 0.001 | 0.0050 | 0.001*** | 0.0251 | < 0.001*** | −20.05 | < 0.001*** |
| 274 | 0.026±0.001 | 0.025±0.001 | 0.0141 | 0.001** | 0.0040 | 0.230 | 1.24 | 0.214 |

$\rho_A$: areal density of high-$Z$ material, $RH_g$: ghost relative radiant fluence, $RH_f$: film relative radiant fluence $D_{KS}$: Kolmogorov–Smirnov effect size, $p$(AD): Anderson–Darling test $p$-value, $p$(HC): binomial test $p$-value, $\theta$: Z-test score, $p$(Z): Z-test $p$-value.

$RH_g$ and $RH_f$: mean values ± 99% CI.

Asterisks signify rejection of null hypothesis at $p$-value:

* < 0.05%,

** < 0.01% and

*** < 0.001%.

with similar chemical composition to melanin. This fact is further supported by the good agreement between experiments and numerical simulations. The physics models in the numerical simulations treat the materials as a mixture of elements or isotopes [26], without taking into account the molecular structure. Thus, the molecular structure of melanin seems to be irrelevant for its shielding capabilities. Moreover, the chemical composition and the pH of the solvent do not affect the shielding capabilities of melanin, as comparable *RD* was measured for synthetic and *S. officinalis* melanin in water and ammonia solvents. The interaction of melanin with Compton recoil electrons has previously been speculated to explain the reduced generation of free radical species from the radiolysis of water [22]. Given that melanin acts also as free radical scavenger [37], identifying the exact source of the observed higher survival rates in biological systems poses a challenging task.

Still melanin may be of interest for shielding applications in space, in an indirect way. Fungal-based biocomposites have been proposed as building and shielding materials for habitat structures on the Moon and Mars [38]. The in-situ production of these materials relies on the resistance of the fungi to the extreme radiation environment of space. Hence, melanized fungi are potential candidates for the production of such biocomposites. Besides, we emphasise that this paper does not address the use of melanin as a possible radioprotector, namely a molecule able to reduce the radiation toxicity and mitigate the health risks from human space flight [4].

The simulation results confirm the spatial arrangement hypothesis; the arrangement of melanin in ghosts may provide increased shielding. Although the arrangement of melanin on the cell membrane will marginally reduce the absorbed dose for highly energetic particles at the nucleus of the cell, the effect will be cumulative for the fungal colony consisting of multiple cells. A lattice of melanin ghosts will feature consistent shielding for the colony cells in an anisotropic radiation environment. Most likely the protective effect of melanin can be attributed to various factors; nonetheless, accumulation of melanin in separate structures inside the cytoplasm instead of a diffused state may have a beneficial effect for shielding. Finally, the discrepancy between our simulations and the Dadachova et al. experiments may have arisen from geometrical variations or differences in the modeling and quantity of the solvents.

The superior performance of composite shields has already been identified by previous studies [36, 39–41] and can be understood from the perspective of dose enhancement effects taking place at the interface of dissimilar materials [42–44]. Most of these studies though approach the problem from another perspective: they either compare the composite shield to aluminum, a commonly used and well-studied material, or they compare different composites materials with one another. Here, we are interested in investigating if there is an optimal geometric arrangement from a radiation shield made of two materials. Our results suggest that spatial arrangement alone is able to reduce the absorbed dose, even up to 50% in comparison to the ideal mixture, just by layering the high-$Z$ material at the incident plane of the radiation. On the other hand, the source of radiation plays an important role; for the $^{90}$Sr source, $h_r$ and $R_{eq}$ play significant role in the dose reduction, whereas for the X-ray source the arrangement itself is the main factor. Moreover, as the shield thickens, which is reflected by an increase in the value of $\rho_A$, the effect of the spatial arrangement seem to diminish, as shown in Table 2. Therefore, for composite shield design, the radiation source and the $\rho_A$ should be taken into account. Although the current study has only examined the reduction in the radiant fluence through the shield, neglecting other design considerations such as manufacturability and cost, it provides an overview on the main design consideration for composite radiation shields.

## Supporting information

**S1 Text. Detailed description of simulation parameters and $\beta$-source selection.** (PDF)

**S1 Fig. Comparison of Compton electron and $^{90}$Sr spectra.** Comparison of Compton electron energy spectrum for the $^{137}$Cs and the $^{90}$Co to the $\beta$-spectrum of $^{90}$Sr. Compton electron spectra were calculated as as described in [45]. (TIF)

**S2 Fig. Spectra used in Geant4 simulations.** (A) Energy spectrum for the $^{90}$Sr source. (B) Energy spectrum for the 40 kVp X-ray source. (TIF)

**S3 Fig. Example of recorded sample and control spectra.** Comparison of the recorded spectrum for the *S. officinalis* and water suspension to its control (water only). (TIF)

**S4 Fig. Simulated $^{90}$Sr spectra for two detector models.** (A) Comparison between the simulated detector spectra, using the source spectrum from [29] (Devaney) or the Geant4 radioactive decay module (Geant4), for a detector that adds the contribution of the secondary particles to the primary and the experimentally recorded spectrum (exp). (B) Simulated spectra for a detector that registers each particle separately. The experimentally recorded spectrum is the same as in panel (A). (TIF)

**S5 Fig. Comparison of different arrangements.** Relative radiant fluence for different spatial arrangements for the melanin-water composite and the X-ray source. The relative film position is marked on the bottom $x$-axis and the equivalent radius for the lattice configurations at the top logarithmic $x$-axis. (TIF)

**S1 Table. Composition of the simulated materials.** Elemental composition used in the numerical simulations for the synthetic and the *S. officinalis* melanins, and the cellulose.
(PDF)

**S2 Table. Correspondence between samples and controls.** Summary of the samples with the corresponding controls and experimental campaigns.
(PDF)

# Acknowledgments

We thank Alessandra Costantino and Michele Muschitiello from the Co-60 Facility (ESTEC, the Netherlands) for the valuable input in the shielding experiments and the handling of the radioactive sources. We thank Alan Dowson from the Life Science Laboratory (ESTEC, the Netherlands) for his support with the sample preparation.

# Author Contributions

**Conceptualization:** Thomas Vasileiou, Leopold Summerer.

**Data curation:** Thomas Vasileiou.

**Formal analysis:** Thomas Vasileiou.

**Investigation:** Thomas Vasileiou.

**Project administration:** Leopold Summerer.

**Software:** Thomas Vasileiou.

**Visualization:** Thomas Vasileiou.

**Writing – original draft:** Thomas Vasileiou, Leopold Summerer.

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
