## [Decision Letter · Decision Letter 0]

14 Jan 2020

PONE-D-19-27509

A biomimetic approach to shielding from ionizing radiation: the case of melanized fungi

PLOS ONE

Dear Dr. Vasileiou,

Thank you for submitting your manuscript to PLOS ONE. After careful consideration, we feel that it has merit but does not fully meet PLOS ONE’s publication criteria as it currently stands. Therefore, we invite you to submit a revised version of the manuscript that addresses the points raised during the review process.

We would appreciate receiving your revised manuscript by Feb 28 2020 11:59PM. To enhance the reproducibility of your results, we recommend that if applicable you deposit your laboratory protocols in protocols.io, where a protocol can be assigned its own identifier (DOI) such that it can be cited independently in the future. For instructions see: http://journals.plos.org/plosone/s/submission-guidelines#loc-laboratory-protocols

We look forward to receiving your revised manuscript.

Kind regards,

Soile Tapio

Academic Editor

PLOS ONE

Journal Requirements:

2. In order to enhance reproducibility, please clarify the origin of the strains used.

3. Our internal editors have looked over your manuscript and determined that it may be within the scope of our Life in Extreme Environments Call for Papers. The Collection will encompass a diverse range of research articles to better understand life and biogeochemistry in extreme environments. Additional information can be found on our announcement page: https://collections.plos.org/s/extreme-environments. If you would like your manuscript to be considered for this collection, please let us know in your cover letter and we will ensure that your paper is treated as if you were responding to this call. If you would prefer to remove your manuscript from collection consideration, please specify this in the cover letter.

Reviewers' comments:

Reviewer's Responses to Questions

**Comments to the Author**

1. Is the manuscript technically sound, and do the data support the conclusions?

Reviewer #1: No

Reviewer #2: Yes

Reviewer #3: Yes

2. Has the statistical analysis been performed appropriately and rigorously? 

Reviewer #1: I Don't Know

Reviewer #2: Yes

Reviewer #3: Yes

3. Have the authors made all data underlying the findings in their manuscript fully available?

Reviewer #1: Yes

Reviewer #2: Yes

Reviewer #3: Yes

4. Is the manuscript presented in an intelligible fashion and written in standard English?

Reviewer #1: Yes

Reviewer #2: Yes

Reviewer #3: Yes

5. Review Comments to the Author

Reviewer #1: The authors raise the possibility of using melanin as a shielding material against space radiation. The argument stems from published research suggesting that the extremely thin layer of melanin in a cell "shields" it from incoming ionizing radiation. To test this, they conduct shielding experiments with melanin and cellulose. They used low energy polyenergetic x-rays and polyenergetic beta particles from Sr-90 to conduct the experiments. The use of polyenergetic beams is not the best choice when conducting shielding studies. Monoenergetic beams are preferred. Furthermore, the type and energy of these radiations is nothing like those encountered in space. High-Z and high-energy particulate (HZE) radiations are encountered in space. These are extremely energetic and can penetrate even thick layers of titanium. Therefore, the thought that a thin layer of melanin has any useful physical shielding characteristics is a stretch. A quick study of stopping powers and attenuation coefficients would have answered this question without conducting experiments and modeling studies. With that in mind, it would be more helpful to focus on the biological mechanisms that may afford protection when melanin content is high.

Reviewer #2: The work presented in the manuscript is very interesting and technically sound. Both the hypothesis about melanin shielding by beta radiation and spatial arrangement have been well presented by the experiments combined with simulation. Experimental results are backed by good statistically approaches.

The work can be published in the journal after the following comment has been addressed.

1. In the figure 3b, the experimental and simulation studies do not agree well in the low energy regime. Authors attribute it to 'spectrum of the 90Sr and neglected detector dynamics'. Authors need to explain why it affects only certain portion of spectrum. Explanation at present is weak.

Reviewer #3: This is an interesting study. The manuscript is written well and I am happy to recommend acceptance of the study. I would like the authors to make a minor change, Lines 310 - 314 ("For the spatial arrangement hypothesis, we used melanin (synthetic, see S1 Table for elemental composition) and water as the high-Z and low-Z materials respectively. We fixed the geometric parameters of the composite shield at A = 3:37 mg cm2 and RV = 0:234. We simulated various configuration for hr from 0 to 1 and Req from 16nm

to 8192nm with the 90Sr and the 40 kVp X-ray source") may be transferred to the methods section..

6. PLOS authors have the option to publish the peer review history of their article (what does this mean?). If published, this will include your full peer review and any attached files.

Reviewer #1: No

Reviewer #2: No

Reviewer #3: No

---

## [Author Response · Author response to Decision Letter 0]

17 Feb 2020

1. Please ensure that your manuscript meets PLOS ONE's style requirements, including those for file naming. The PLOS ONE style templates can be found at [URL_removed] and [URL_removed].

Response: We have reviewed and updated the manuscript, so the style conforms to the given guidelines. We have also checked the names and the addresses of the authors for spellings or typos.

2. In order to enhance reproducibility, please clarify the origin of the strains used.

Response: In our experiments we have not used any living organism or cell lines. Naturally synthesized melanin (like for example the one from S. officinalis) and cellulose was purchased already extracted and purified. In all such cases, we indicate the supplier and the product number in the “Chemicals and sample preparation” subsection.

3. Our internal editors have looked over your manuscript and determined that it may be within the scope of our Life in Extreme Environments Call for Papers. The Collection will encompass a diverse range of research articles to better understand life and biogeochemistry in extreme environments. Additional information can be found on our announcement page: [URL_removed]. If you would like your manuscript to be considered for this collection, please let us know in your cover letter and we will ensure that your paper is treated as if you were responding to this call. If you would prefer to remove your manuscript from collection consideration, please specify this in the cover letter.

Response: Thank you for pointing out this possibility. We are indeed interested in including the manuscript in the Life in Extreme Environments collection.

Response: We adhere to our previous Data Availability statement. The dataset and the code used in the manuscript are deposited in Zenodo, DOI: 10.5281/zenodo.3667494.

Reviewer #1: The authors raise the possibility of using melanin as a shielding material against space radiation. The argument stems from published research suggesting that the extremely thin layer of melanin in a cell "shields" it from incoming ionizing radiation. To test this, they conduct shielding experiments with melanin and cellulose. They used low energy polyenergetic x-rays and polyenergetic beta particles from Sr-90 to conduct the experiments. The use of polyenergetic beams is not the best choice when conducting shielding studies. Monoenergetic beams are preferred. Furthermore, the type and energy of these radiations is nothing like those encountered in space. High-Z and high-energy particulate (HZE) radiations are encountered in space. These are extremely energetic and can penetrate even thick layers of titanium. Therefore, the thought that a thin layer of melanin has any useful physical shielding characteristics is a stretch. A quick study of stopping powers and attenuation coefficients would have answered this question without conducting experiments and modeling studies. With that in mind, it would be more helpful to focus on the biological mechanisms that may afford protection when melanin content is high.

Response: The manuscript discuss reported mechanisms to mitigate the effects of radiation damage, by investigating how melanized fungi manage to survive in high-radiation environments. Indeed, we fully agree with the reviewer that a thin film of melanin would be inadequate to protect against HZE. We make this also clear in the “Discussion” section of the manuscript, where we doubt that melanin is able to provide improved shielding even from the significantly less energetic β-radiation that we have employed in our experiments:

l.368-370: “With respect to interaction of melanin with β-radiation, our experiments indicate that melanin does not exhibit improved shielding; the recorded RD for melanin was comparable to cellulose, a substance with similar chemical composition to melanin.”

We want to point out that the manuscript does investigate one of the speculated biological mechanisms for the experimentally reported and published protective property of melanin; the physical interaction with the secondary β-particles. Fully consistent with published literature on the topic, the word “shielding” is used throughout the manuscript to indicate physical interaction of melanin with ionizing radiation, in contrast to biochemical, even if the melanin lies inside a living organism (a terminology that was not introduced by the authors). To make this point more clear we have added the following sentence in the Introduction:

l.36-37 : “The word shielding is used here to indicate the physical interaction between radiation and melanin, even if the latter resides inside the cell.”

Until today, it is unclear if this shielding plays a significant role, mainly because the vast majority of the previous experiments used model organism (animal or fungi cultures), where biochemical effects may mask physical and vice versa; by clearly separating the two, we hope that our experiment sheds some light on the matter. 

Moreover, the shielding properties of melanin have caught the attention of the space community, which puts money and resources to pursue such potential; for example with the $200.000 project “Protecting Astronauts from Space Radiation-Induced Carcinogenesis and Central Nervous System Damage with Melanin-Containing Food and Materials” from the Canadian Space Agency [1] or the collaboration of John Hopkins University and NASA to send melanin samples to ISS [2,3].

For the aforementioned reason, the selected sources for our experiments do not intend to simulate the space radiation environment but to reproduce energy spectra from previous experiments; 90Sr source imitates the Compton electron spectrum of 60Co and 137Cs sources (detailed Compton spectra derivation can be found in S1 Text – “Selection of radioactive source”). The X-ray spectrum is similar to previous experiments that connected melanin ghosts to increased attenuation of X-rays, as described in the “Spatial arrangement” subsection.

Further, the manuscript also proposes ways to improve radiation shielding materials; while it is trivial to estimate the shielding characteristics from stopping powers and attenuation coefficients for a single material (as the reviewer correctly points out), the same calculation for a composite material is a much more challenging and complex task. Our results suggest that arranging two material into different configurations may substantially change the shielding characteristics. Therefore, improved shielding effectiveness can be achieved for reduced shield mass, which is of interest in space flight. As we state in the manuscript, the superior shielding of composite materials has already been identified by the community, but to our knowledge, rationalization of the composite configurations or the use of micro- or nano-particle have not been discussed before in the literature.

Lastly, shielding experiments with monoenergetic beams have the value to characterize in detail shielding materials. Polyenergetic beams are able to resolve the magnitude of the attenuation of a given material and useful as a mean to assess if a monoenergetic experimental campaign is justified. As explained in the paper, we did not detect any significant effect of melanin in our polyenergetic experiments that support further investigation and we are confident that monoenergetic experiments will lead to the same conclusion.

References

[1] "Grants awarded under the FAST 2017 Announcement of Opportunity," 14 March 2019. [Online]. Available: https://www.asc-csa.gc.ca/eng/funding-programs/programs/fast/grants-awarded-fast-ao-2017.asp. [Accessed 30 January 2020].

[2] "Biological pigment that acts as nature's sunscreen set for space journey," Hub - Johns Hopkins University, [Online]. Available: https://hub.jhu.edu/2019/11/01/melanin-space-study/. [Accessed 13 February 2020].

[3] "NASA Tests Melanin-Based Radiation Blocker for Astronauts," Machine Design, [Online]. Available: https://www.machinedesign.com/materials/article/21838319/nasa-tests-melaninbased-radiation-blocker-for-astronauts. [Accessed 5 February 2020].

Reviewer #2: The work presented in the manuscript is very interesting and technically sound. Both the hypothesis about melanin shielding by beta radiation and spatial arrangement have been well presented by the experiments combined with simulation. Experimental results are backed by good statistically approaches.

The work can be published in the journal after the following comment has been addressed.

1. In the figure 3b, the experimental and simulation studies do not agree well in the low energy regime. Authors attribute it to 'spectrum of the 90Sr and neglected detector dynamics'. Authors need to explain why it affects only certain portion of spectrum. Explanation at present is weak.

Response: We have updated the manuscript in order to better explain the observed discrepancy between the 90Sr spectra recorded during the experiments and simulation. For calibration, we had acquired spectra of the 90Sr source without the melanin samples at the beginning of each experimental campaign. We simulated the same conditions in Geant4, namely the source and the detector (without the melanin sample) and we observed the same discrepancy between experiment and simulation; the experimental spectrum has more content in the low energy region. These result are shown in Figure S4A, with the experiment labeled as “exp” and the simulations are “Geant4”. Therefore, we feel safe to conclude that this effect is not connect to the presence of melanin, the solvent or the container.

The most likely reason for the observed effect is the simplified model we used for the detector. We remind to the reviewer that for the simulations we have used the spectrum taken from [1] instead of fitting the recorded source spectrum to the observed one from the experiments, in an attempt to avoid any systematic errors. Further, we have modelled the detector using the following assumption: the detector registers the total deposited energy of the primary particle, that is the sum of the deposited energy from the primary and all the secondary particles. This assumptions inaccurately captures the exact detector dynamics, which is determined by the electronics and the signal processing algorithms that are applied to the voltage pulses induced in the crystal by the incoming particles. During the processing step, pulses can be merged or rejected, as stated in the manual [2]. To further investigate if these neglected dynamics are able to produce the magnitude of the observed shift, we modelled the detector that is able to record any secondary particle as a separate event, independent from the primary that created it. As before, we used the spectrum from [1] and the radioactive decay module of Geant4. The latter simulates an immobile 90Sr nuclei and tracks all the intermediate nuclear reactions and products, making it computationally more demanding. It is expected that this detector model will shift the content of the histogram towards lower energies. The comparison of the two detector models is shown in Figure S4. Firstly, both the spectrum from Devaney and the radioactive decay module of Geant4 produce similar results. More importantly, the neglected detector dynamics are able to produce the observed effect between simulation and experiment, with the real detector response most likely lying somewhere between these two extreme cases.

The way we have selected to model the detector seems to sacrifice some of the accuracy of the final spectrum, but it captures the overall trend. Moreover, the dose calculation is not affected by the detector modelling, since in the simulations the production and tracking of the secondary particles do not depend on the way they are binned in the histogram. We only expect a minor effect on the estimation of the dose variance, but we remark that we never use statistical comparisons that mix simulations and experiments. The implemented detector model is justified by the tradeoff between accuracy versus complexity and execution time.

The previous discussion and Figure S4 have been included to the supplementary materials. The text in manuscript has be altered to direct the reader to the added SI section. The code that produce the discussed simulations and Figure S4 has been added to the repository.

References

[1] J. J. Devaney, "Beta spectra of ⁹⁰Sr and ⁹⁰Y," Los Alamos National Lab, 1985.

[2] CUBE527 Spectrometer series - User Manual, Radeberg: GBS-Elektronik GmbH, 2018. 

Reviewer #3: This is an interesting study. The manuscript is written well and I am happy to recommend acceptance of the study. I would like the authors to make a minor change, Lines 310 - 314 ("For the spatial arrangement hypothesis, we used melanin (synthetic, see S1 Table for elemental composition) and water as the high-Z and low-Z materials respectively. We fixed the geometric parameters of the composite shield at A = 3:37 mg cm2 and RV = 0:234. We simulated various configuration for hr from 0 to 1 and Req from 16nm

to 8192nm with the 90Sr and the 40 kVp X-ray source") may be transferred to the methods section.

Response: We have moved the specific text to the methods section, as proposed by the reviewer.

---

## [Editor Report · Decision Letter 1]

19 Feb 2020

A biomimetic approach to shielding from ionizing radiation: the case of melanized fungi

PONE-D-19-27509R1

Dear Dr. Vasileiou,

We are pleased to inform you that your manuscript has been judged scientifically suitable for publication and will be formally accepted for publication once it complies with all outstanding technical requirements.

With kind regards,

Soile Tapio

Academic Editor

PLOS ONE
---

## [Editor Report · Acceptance letter]

26 Feb 2020

PONE-D-19-27509R1 

A biomimetic approach to shielding from ionizing radiation: the case of melanized fungi 

Dear Dr. Vasileiou:

I am pleased to inform you that your manuscript has been deemed suitable for publication in PLOS ONE. Congratulations! Your manuscript is now with our production department. 

With kind regards,

on behalf of

Dr. Soile Tapio 

Academic Editor

PLOS ONE